# Proton Beam Therapy for Treating Patients with Hepatocellular Carcinoma with Major Portal Vein Tumor Invasion: A Single Center Retrospective Study

**DOI:** 10.3390/cancers16112050

**Published:** 2024-05-29

**Authors:** Toshiki Ishida, Masashi Mizumoto, Takashi Saito, Toshiyuki Okumura, Kosei Miura, Hirokazu Makishima, Takashi Iizumi, Haruko Numajiri, Keiichiro Baba, Motohiro Murakami, Masatoshi Nakamura, Kei Nakai, Hideyuki Sakurai

**Affiliations:** 1Department of Radiation Oncology, University of Tsukuba Hospital, 2-1-1 Tsukuba, Ibaraki 305-8576, Japan; tishida@pmrc.tsukuba.ac.jp (T.I.); saitoh@pmrc.tsukuba.ac.jp (T.S.); okumura@pmrc.tsukuba.ac.jp (T.O.); kousei951378246@gmail.com (K.M.); hmakishima@pmrc.tsukuba.ac.jp (H.M.); iizumi@pmrc.tsukuba.ac.jp (T.I.); haruko@pmrc.tsukuba.ac.jp (H.N.); baba@pmrc.tsukuba.ac.jp (K.B.); murakami@pmrc.tsukuba.ac.jp (M.M.); nakamura@pmrc.tsukuba.ac.jp (M.N.); knakai@pmrc.tsukuba.ac.jp (K.N.); hsakurai@pmrc.tsukuba.ac.jp (H.S.); 2Department of Radiation Oncology, Ibaraki Prefectural Central Hospital, Ibaraki 309-1703, Japan; 3Department of Radiation Oncology, JCHO Tokyo Shinjuku Medical Center, Tokyo 162-8543, Japan

**Keywords:** hepatocellular carcinoma, proton beam therapy, retrospective, radiotherapy

## Abstract

**Simple Summary:**

Hepatocellular carcinoma (HCC) is a life-threatening disease of the liver. Patients who also have a blockage of the portal vein, which takes blood into the liver, are at particular risk for death. This condition is known as portal vein tumor thrombosis (PVTT). Proton beam therapy (PBT) is an excellent treatment option for tumors because it allows the tumor to be irradiated while avoiding radiation effects on normal tissue. In this study, we found that the long-term outcomes in patients with HCC with advanced PVTT (Vp3 or Vp4) were improved by treatment with PBT. In particular, the median survival time after was >20 months in patients treated with PBT for cure of the disease. These results are better than those with other therapies and suggest that PBT gives a survival benefit in these cases. There were also very few adverse events, indicating that PBT is a safe method.

**Abstract:**

Hepatocellular carcinoma (HCC) with portal vein tumor thrombosis (PVTT) has a poor prognosis and is generally not indicated for surgery. Proton beam therapy (PBT) may offer an alternative treatment. In this study, long-term outcomes were examined in 116 patients (median age 66 years, 100 males) with HCC with advanced PVTT (Vp3 or Vp4) who received PBT from April 2008 to March 2018. Of these patients, 63 received PBT as definitive treatment and 53 as palliative treatment. The representative dose was 72.6 Gy (RBE) in 22 fractions. Eight patients died in follow-up, including 72 due to tumor progression. The 5-year overall survival (OS) rate was 18.0% (95% CI 9.8–26.2%) and the 5-year local control (LC) rate was 86.1% (74.9–97.3%). In multivariate analyses, performance status and treatment strategy were significantly associated with OS. The median follow-up period for survivors with definitive treatment was 33.5 (2–129) months, and the 5-year OS rate was 25.1% (12.9–37.3%) in these cases. The median survival time after definitive irradiation was >20 months. The 5-year OS rate was 9.1% (0–19.7%) for palliative irradiation. These results compare favorably with those of other therapies and suggest that PBT is a useful option for cases of HCC with advanced PVTT that cannot undergo surgery, with an expected survival benefit and good local control. Determining the optimal indication for this treatment is a future challenge.

## 1. Introduction

Hepatocellular carcinoma (HCC) is the most common primary malignancy of the liver [1]. HCC occurs most frequently in individuals with chronic liver disease. Hepatitis B or C infection is the most common cause. Therefore, HCC is prevalent in East Asian countries and sub-Saharan Africa where HBV infection is widespread and is also increasing in the United States and Western Europe. Alcoholic liver disease and non-alcoholic fatty liver disease (NAFLD) are also known risks for HCC [2]. In Japan, HCC is the fifth leading cause of cancer-related deaths. Treatment of HCC generally includes surgery, transarterial chemoembolization (TACE), percutaneous ethanol injection (PEI), percutaneous microwave coagulation (PMC), and radiofrequency ablation (RFA) [3,4,5,6,7,8,9].

The prognosis of HCC with portal vein tumor thrombosis (PVTT) is particularly poor, among other factors such as temporal and spatial multiplicity, cirrhosis, and vascular invasion [10]. HCC with PVTT is generally not indicated for surgery or transplantation, and TACE cannot be used. In the Barcelona Clinic Liver Cancer (BCLC) Staging System, the standard of care for HCC with PVTT is the molecularly targeted agent sorafenib [11]. However, the results are still poor, with survival rates of 3.1–10 months. Radiotherapy is not included as a treatment option in the BCLC algorithm or in liver cancer practice guidelines in Japan, but evidence for the efficacy of SBRT is gradually emerging [12,13].

Proton beam therapy (PBT) is not included in HCC treatment guidelines but has been reported to give good local control that is like standard local treatments such as surgery and RFA [14] and has been suggested to be safe for HCC with PVTT [15]. However, there are few reports on the long-term efficacy and safety of PBT for HCC with PVTT. Thus, in this study, we analyzed the long-term clinical outcomes of PBT for cases of HCC with advanced PVTT (Vp3 or Vp4) treated at our center.

## 2. Materials and Methods

### 2.1. Patients

A retrospective investigation was performed on 116 patients with HCC with advanced PVTT (Vp3 or Vp4) who received PBT at the University of Tsukuba Hospital from April 2008 to March 2018. The patient and tumor characteristics are shown in Table 1. The median age was 66 (range: 27–88) years, and 100 cases were male and 16 were female. The ECOG (Eastern Cooperative Oncology Group) performance status (PS) was 0, 1, 2, and 3 in 71, 39, 4, and 2 cases, respectively. The median tumor size was 60 (range: 10–200) mm. The Child–Pugh classification was A, B, and C in 87, 29, and 0 cases, respectively. Of the 116 patients, 31 had received other therapy prior to PBT, and 63 received PBT as definitive treatment, which was defined as irradiation of all active lesions, including the primary tumor and PVTT plus all other intrahepatic lesions and lymph node metastases if present and a minimum tumor dose ≥ 50 Gy. Even in cases where previous lesions have been controlled long-term by prior treatments and new lesions with PVTT have appeared, they are defined as definitive treatment if the entire lesion is included in the irradiated area. PVTT was classified as Vp3 (tumor extension to a primary branch of the portal vein) in 63 cases and Vp4 (tumor invasion into the main trunk of the portal vein or the other primary branch) in 53 cases.

### 2.2. Proton Beam Therapy

PBT was performed under the following conditions. A fixture was made and planned CT was taken in 5-mm slices under respiratory synchronization. The target volume included the primary tumor and the portal vein tumor plug, and if liver function permitted, the entire segment or segments including the primary tumor and PVTT were including the primary tumor was considered as the clinical target volume. Such cases in which gross lesions could be included in the irradiation field were defined as definitive treatment. In a case with extensive tumor progression or distant metastasis, only the tumor plug could be targeted. We defined it as palliative treatment. The dose constraint was aimed at 54 Gy (EQD2 α/β = 3) for the stomach, duodenum, and colon. The dose fraction is determined by the location of the lesion. The representative dose fraction was set at 72.6 Gy (RBE) in 22 fractions when the tumor was located at the porta hepatis. When the lesion is close to the gastrointestinal tract, the dose is reduced to 2 Gy (RBE) per fraction in consideration of the tolerable dose of the gastrointestinal tract. In addition, the dose fractionation may be adjusted according to other factors such as outpatient or not, patient’s health is adequate or not. A total of 60.0 Gy (RBE) in 15 fractions is the second most used dose fractionation, and there are 5 cases with less than 60.0 Gy (RBE) in palliative treatment. A synchrotron proton accelerator of 155–250 MeV was used, with beam modulation using the passive scattering method. Image-guided irradiation (gold marker, fluoroscopy) and respiratory-synchronized irradiation were used during treatment. Adverse events were determined according to the Common Terminology Criteria for Adverse Events v.4.0 [16].

### 2.3. Statistical Analysis

Overall survival (OS) and local control (LC) were calculated using the Kaplan–Meier method with SPSS (v.27, Chicago, IL, USA). OS and LC rates were calculated using the date of initiation of PBT as the starting date, and local recurrence was defined as the date the lesion that had been irradiated showed apparent growth on diagnostic images. Pretreatment prognostic factors were examined by univariate analysis using a log-rank test. Multivariate analysis was performed using a Cox proportional hazards model. Prognostic factors used in the calculations included pretreatment (yes vs. no), Child–Pugh score (<7, class A vs. ≥7 class B, C), treatment strategy (definitive irradiation including all lesions vs. palliative irradiation for PVTT only), dose (≥72.6 vs. <72.6 Gy (RBE)), location of tumor plug (Vp3 vs. Vp4), PS (0 vs. ≥1), PIVKA II (≥1000 vs. <1000), and AFP (≥100 vs. <100).

## 3. Results

### 3.1. Overall Survival and Local Control

Of the 116 HCC patients with advanced PVTT (Vp3 or Vp4) who received PBT, 88 had died as of the last follow-up date due to tumor progression (*n* = 72) and other causes (*n* = 14), including ruptured esophageal varices (*n* = 4), liver failure (*n* = 3), other cancer (*n* = 2), respiratory disease (*n* = 2), stroke (*n* = 1), traffic accident (*n* = 1), complication after liver transplantation (*n* = 1), and unknown causes (*n* = 2). The median follow-up period for survivors of cases treated with definitive irradiation was 33.5 (2–129) months. In all cases, the 1-, 2-, 3-, 4-, and 5-year OS rates were 49.9% (95% CI 40.5–59.3), 34.6% (25.4–43.8%), 29.8% (20.8–38.8%), 23.4% (14.8–32.0%), and 18.0% (9.8–26.2%); and the respective LC rates were 95.1% (90.2–100%), 90.6% (83.0–98.2%), 90.6% (83.0–98.2%), 86.1% (74.9–97.3%), and 86.1% (74.9–97.3%). The median survival time (MST) was 11.0 (95% CI 7.1–14.9) months. The OS and LC rates are shown in Figure 1. In multivariate analyses, PS and treatment strategy were significantly associated with OS, but no significant factors were associated with LC (Table 2).

The 1-, 3-, and 5-year OS rates were 64.8% (52.6–77.0%), 40.3% (27.4–53.2%), and 25.1% (12.9–37.3%) in cases treated with definitive PBT, and 29.9% (17.2–42.6%), 18.3% (7.1–29.5%), and 9.1% (0–19.7%) for palliative PBT cases (*p* = 0.001). The respective MSTs were 24.0 (13.2–34.8) and 8.0 (5.7–10.3) months. The 1-, 3-, and 5-year LC rates were 95.8% (90.1–100%), 90.1% (80.7–99.5%), and 85.1% (72.2–98.0%) after definitive PBT, and 94.4% (86.6–100%), 94.4% (86.6–100%), and 94.4% (86.6–100%) after palliative PBT (*p* = 0.999). OS and LC rates in definitive and palliative PBT cases are shown in Figure 2.

### 3.2. Analysis Based on PVTT

Regarding progression of PVTT (Vp3 vs. Vp4), the 1-, 3-, and 5-year OS rates were 57.9% (45.6–70.2%), 36.4% (23.9–48.9%), and 19.2% (7.8–30.6%) in Vp3 cases, and 37.5% (23.6–51.4%), 21.6% (9.1–34.1%), and 16.2% (4.8–27.6%) in Vp4 cases (*p* = 0.125). The 1-, 3-, and 5-year LC rates were 95.5% (89.4–100%), 92.2% (83.6–100%), and 85.1% (69.6–100%) in Vp3 cases, and 94.9% (87.9–100%), 88.5% (74.8–100%), and 88.5% (74.8–100%) in Vp4 cases (*p* = 0.999). OS and LC rates by progression of PVTT are shown in Figure 3.

OS and LC rates of Vp3 and Vp4 cases were also compared in cases that underwent definitive PBT. The 1-, 3-, and 5-year OS rates were 67.2% (52.0–81.9%), 45.6% (29.7–61.5%), and 25.6% (10.5–40.7%) for Vp3/definitive PBT cases, and 60.1% (38.3–81.9%), 30.0% (8.4–51.6%), and 24.0% (3.8–44.2%) for Vp4/definitive PBT cases (*p* = 0.408). The respective MSTs were 23.0 (0.0–50.3) and 26.0 (5.7–46.3) months. The 1-, 3-, and 5-year LC rates were 94.0% (86.0–100%), 90.1% (79.3–100%), and 83.1% (66.6–99.6%) for Vp3/definitive PBT cases and 100%, 90.0% (71.4–100%), and 90.0% (71.4–100%) for Vp4/definitive PBT cases (*p* = 0.602). OS and LC rates by progression of PVTT in definitive PBT cases are shown in Figure 4.

### 3.3. Recurrence and Late Adverse Events

The first recurrences in all patients (*n* = 116) were within the irradiated field (*n* = 7), intrahepatic recurrence outside the irradiated field (*n* = 70), lymph node recurrence (*n* = 7), and distant metastasis recurrence (*n* = 25), with 5, 45, 5, and 13 in definitive irradiation cases (*n* = 63), and 2, 25, 2, and 12 in palliative cases (*n* = 53), respectively.

PBT-related late adverse events of Grade (Gr) 2 or higher in all patients (*n* = 116) were radiation dermatitis (*n* = 3, all Gr 2), rib fracture (*n* = 6, all Gr 2), gastrointestinal ulcer/stenosis (*n* = 4, 2 Gr 2, 2 Gr 3), and cholecystitis (*n* = 1, Gr 3); with 2, 5, 1 (Gr 3), and 1 in definitive irradiation cases, and 1, 1, 3 (2 Gr2, 1 Gr3), and 0 in palliative cases, respectively.

## 4. Discussion

HCC is more common in males, with a male to female ratio of 1:2.4 in the worldwide. Though the ratio is similar in Japan, the proportion of males in this study was higher than that. Since there is no difference between men and women in patient selection criteria, it is difficult to explain why the proportion of men is so high. Many cases in this study were treated without insurance, and it is possible that women were less likely to use PBT due to the economic disparity between men and women. Therefore, this ratio may change as public insurance coverage becomes available for hepatocellular carcinoma of 4 cm or larger starting in April 2022 in Japan. The Japan Liver Cancer Association summarizes pathological progression with HCC by gender and shows that the proportion of males increases as the degree of progression in vascular invasion.

The prognosis for advanced PVTT is poor, with a median untreated survival of only 5–10 months [17,18]. Surgery and radiofrequency are often difficult to perform for locally advanced PVTT, and cases treated with molecularly targeted drugs and cytotoxic chemotherapy have poor prognoses [19,20,21]. The Efficacies of hepatic arterial infusion chemotherapy (HAIC) and transarterial radioembolization (TARE) for PVTT are expected, but they have not been established as standard treatment modalities [21,22,23]. Radiation therapy for HCC is often used in cases where surgery or RFA is not feasible or for palliation of symptoms [24,25,26]. 3D-CRT is often used in combination with TACE, while SBRT and PBT are used as local treatments regardless of combination therapy [24,25,26]. A meta-analysis of radiation therapy for HCC reported that PBT and SBRT had nearly equivalent outcomes and that 3DCRT was inferior in survival [27,28]. SBRT is mainly used for tumors that are 3–4 cm or smaller, whereas PBT can be performed regardless of the tumor size [29,30].

There have been several reports of radiation therapy for PVTT, with the median OS found to be about 12 months with 3DCRT and SBRT, and more than 20 months with PBT (Table 3) [15,31,32,33,34,35,36,37,38,39,40]. Accurate comparisons are difficult due to the small number of reports, but PBT seems to have better results in radiation therapy for PVTT. This may be due to the large tumor size in PVTT cases and the possibility of low hepatic reserve capacity due to reduced hepatic blood flow caused by PVTT. The general outcome of PBT for HCC is a 5-year local control rate of about 80–90%, and the main form of recurrence is intrahepatic recurrence in the irradiated area [41,42]. In Japan, PBT is performed according to tumor localization and dose division according to the unified treatment policy established by JASTRO.

In this study, more than 100 cases were accumulated compared to previous reports from our own institution, allowing us to compare the difference in results between Vp3 and Vp4. In the present analysis, local control was similar regardless of definitive or palliative irradiation (Vp3 and Vp4), but the OS rate was clearly worse in palliative cases. This indicates that outcomes depend on the patient’s background, as found previously [43,44]. Multivariate analysis also showed a trend for a relationship between PS and liver function with OS. The MST of advanced PVTT cases treated with definitive irradiation was more than 20 months, which is comparable to previous results and a good therapeutic outcome [15,31]. These results compare favorably with those with molecular-targeted drugs and TACE, and suggest that PBT is a useful treatment option for PVTT cases that cannot undergo surgery or RFA [45]. On the other hand, the MST for palliative cases was about 8 months, but this appears acceptable compared to no treatment or use of molecular-targeted drugs alone [46]. Although the indication criteria for PBT in palliative cases are unclear, a survival benefit may be expected if advanced PVTT is likely to have a nodal effect on the prognosis. In this study, 60.0 Gy (RBE) in 15 fractions was often chosen, but dose fractionation with a shorter treatment period may be a better option.

There have been few reports of adverse events related to blood vessels and bile ducts in the hilar region due to PBT. In this study, most patients received high doses of about 72.6 Gy (RBE) in 22 fractions to the porta hepatis, but there were no adverse events that were problematic for the blood vessels or bile ducts.

There have been an increasing number of reports of the use of lenvatinib in combination with SBRT or TACE for PVTT [13,47]. Given that most recurrence sites after PBT in this study were outside the irradiated area, it is important to administer the irradiation necessary to limit death from local progression in as short a time as safely possible. Moreover, it is likely that the use of PBT in combination with agents such as lenvatinib will improve treatment outcomes. In recent years, several clinical trials have been conducted and reported on the combination of immune checkpoint inhibitors and PBT for HCC [48,49]. In Japan, several clinical trials and prospective studies of PBT for HCC have been conducted. For example, Phase Ib trial of durvalumab plus tremelimumab in combination with particle therapy in advanced hepatocellular carcinoma patients with macrovascular invasion: DEPARTURE trial and a non-randomized controlled study comparing proton beam therapy and hepatectomy for resectable hepatocellular carcinoma: SPRING study has been conducted. In addition, based on the results of this study, we plan to prospective observational study to select the irradiation method according to the patient background in our hospital.

## 5. Conclusions

Proton beam therapy achieved good local control of HCC with advanced PVTT. While definitive irradiation offers a good prognosis, palliative irradiation has problems in controlling the lesions outside the irradiated area. Determining the optimal indication and optimal combination of other therapies is a future challenge.

## Figures and Tables

**Figure 1 cancers-16-02050-f001:**
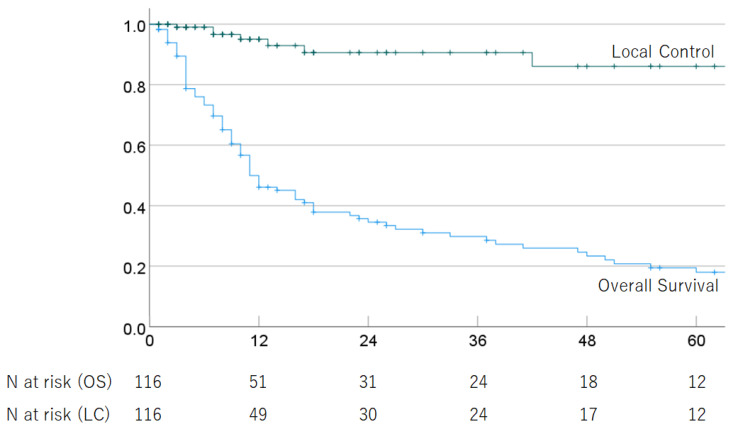
Overall survival rate and local control rate in all patients.

**Figure 2 cancers-16-02050-f002:**
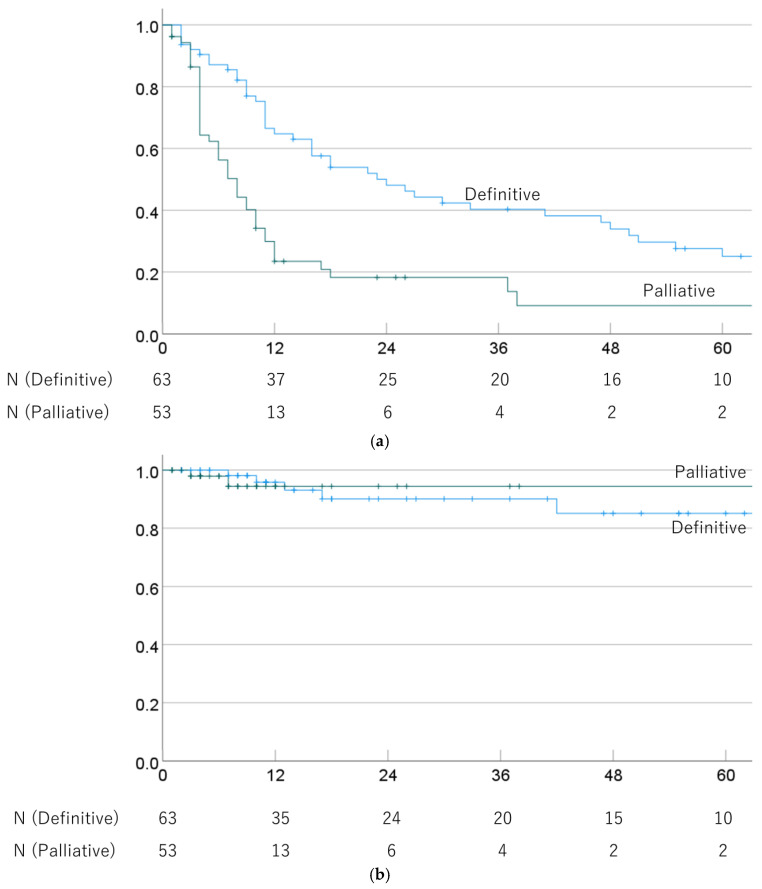
(**a**) Overall survival rate and (**b**) local control rate in cases treated with definitive and palliative PBT.

**Figure 3 cancers-16-02050-f003:**
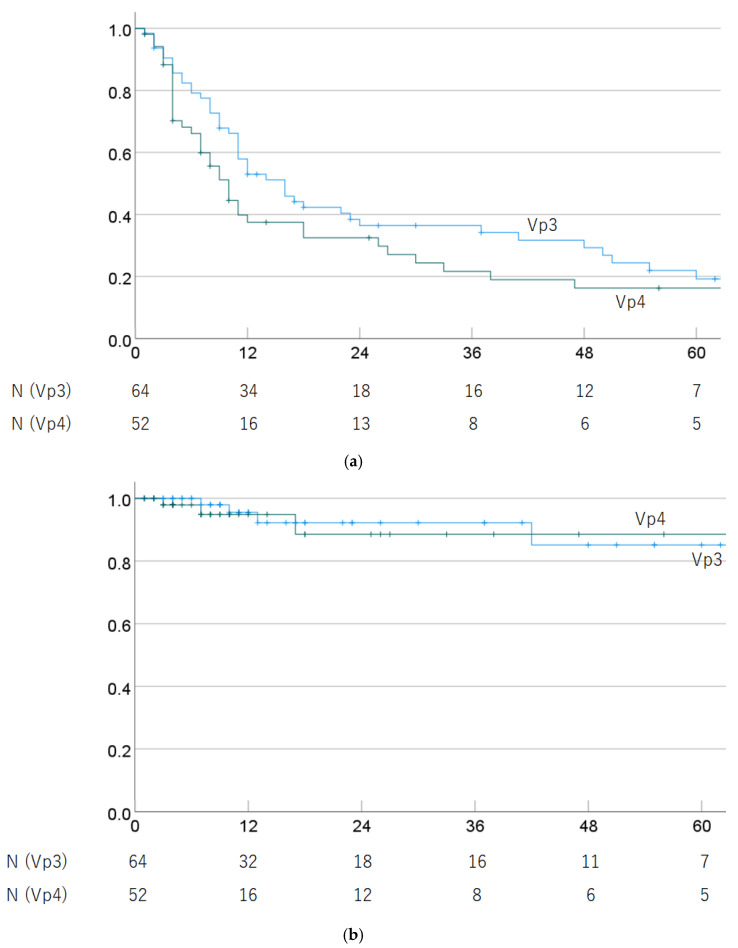
(**a**) Overall survival rate and (**b**) local control rate by progression of PVTT (Vp3 vs. Vp4).

**Figure 4 cancers-16-02050-f004:**
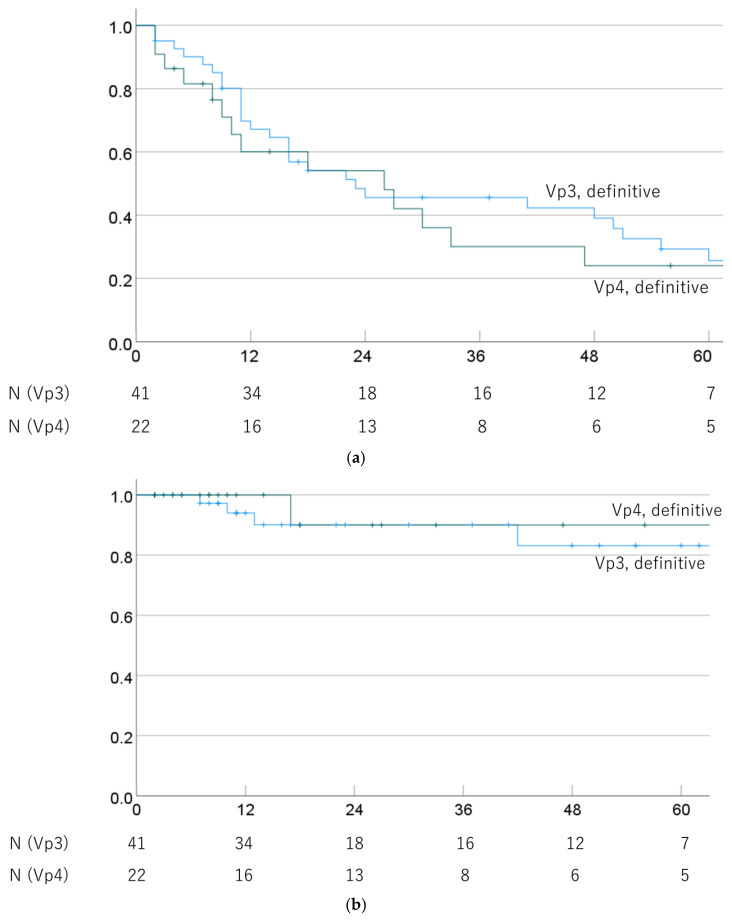
(**a**) Overall survival rate and (**b**) local control rate by progression of PVTT (Vp3 vs. Vp4) in cases treated with definitive PBT.

**Table 1 cancers-16-02050-t001:** Patient and tumor characteristics.

Characteristics	Number	%
Age	27–88	66 (median)
Gender		
Male	100	
Female	16	
ECOG performance status		
0	71	
1–3	45	
Tumor size (mm)		
	10–200	60 (median)
Child-Pugh class		
A	87	
B	29	
C	0	
Prior treatment		
Yes	31	
No	85	
Portal vein tumor thrombosis		
Vp3	64	
Vp4	52	
Definitive treatment		
Yes	63	
No	53	
Dose fractionation		
72.6 Gy (RBE) in 22 fractions	73	
60.0 Gy (RBE) in 15 fractions	15	
74.0 Gy (RBE) in 37 fractions	8	
66.0 Gy (RBE) in 20 fractions	5	
Others	15	

ECOG, Eastern Cooperative Oncology Group.

**Table 2 cancers-16-02050-t002:** Multivariate analysis of potential predictive factors for overall survival and local recurrence.

Factors	PT Number	3-Year (%)	5-Year (%)	*p* Value	HR	95% CI
Overall survival						
Performance status				0.027	1.65	1.06–2.56
0	71	34	22			
1–3	45	23	11			
Prior treatment				0.288	0.77	0.48–1.24
Yes	31	24	17			
No	85	33	18			
Child-Pugh class				0.097	1.51	0.94–2.44
A	85	35	24			
B/C	31	17	0			
Treatment strategy				0.001	2.25	1.39–3.64
Definitive	63	40	25			
Palliative	53	18	9			
Total dose				0.520	0.85	0.52–1.40
≥72.6 Gy (RBE)	83	35	20			
<72.6 Gy (RBE)	33	16	12			
PVTT				0.632	1.12	0.71–1.74
Vp3	64	36	19			
Vp4	52	22	16			
PIVKA-II				0.068	1.53	0.97–2.41
<1000	64	38	19			
≧1000	52	20	17			
Local control						
Performance status				0.42	0.400	0.04–3.67
0	71	87	87			
1–3	45	100	83			
Prior treatment				0.83	1.210	0.22–6.79
Yes	31	84	84			
No	85	93	87			
Child-Pugh class				0.97	0	0
A	85	88	84			
B/C	31	100	100			
Treatment strategy				0.73	0.69	0.09–5.61
Definitive	63	90	86			
Palliative	53	95	95			
Total dose				0.72	0.68	0.08–5.55
≥72.6 Gy (RBE)	83	91	86			
<72.6 Gy (RBE)	33	91	89			
PVTT				0.95	1.06	0.21–5.28
Vp3	64	93	86			
Vp4	52	89	89			
PIVKA-II				0.06	5.14	0.97–27.3
<1000	64	96	96			
≥1000	52	78	65			

PIVKA-II = protein induced by vitamin K absence or antagonist-II, HR = hazard ratio, CI = confidence interval.

**Table 3 cancers-16-02050-t003:** Outcomes of radiotherapy for PVTT.

Author	Year	*n*	Institution	Age	Modality	Palliative	Size	MST	OS 1y	OS 2y	OS 3y
This study	2023	63	Tsukuba	66	Proton	0	60	24.0	64.8	-	40.3
Lee [31]	2014	27	NCC	55	Proton	5	70	13.2	55.6	33.3	-
Sugahara [15]	2009	35	Tsukuba	63	Proton	0	60	22	61	48	40
Shui [32]	2018	70	Affliated	70	SBRT	N/A	-	10	40	-	-
Matsuo [33]	2016	43	Kobe	70	SBRT	4	31	12	49	-	-
Matsuo [33]	2016	54	Kobe	69	3DCRT	7	32	6.5	38	-	-
Wang [34]	2016	56	Multi	50	3DCRT	0	-	8.9	38	19	17
Lu [35]	2015	30	PLA	59	3DCRT	0	-	13	62.4	20.8	-
Kim [36]	2015	102	Asan	-	3DCRT	N/A	-	11.4	-	-	-
Han [37]	2008	40	Yonsei	50	3DCRT	0	-	13.1	57.6	32.2	24.1
Toya [38]	2007	38	Kumamoto	67	3DCRT	0	40	9.6	39.4	19	0
Kim [39]	2005	59	NCC	57	3DCRT	N/A	110	10.7	40.7	20.7	-
Huang [40]	2001	41	Kaohsiung	-	3DCRT	N/A	100	10	40	10	3

MST: median survival time, OS: overall survival, SBRT: stereotactic body radiation therapy, NCC: National Cancer Center.

## Data Availability

All data are included in the paper.

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
