# Peer review of "Proton Beam Therapy for Treating Patients with Hepatocellular Carcinoma with Major Portal Vein Tumor Invasion: A Single Center Retrospective Study"

_cancers, 2024, doi:10.3390/cancers16112050_

Round 1
Reviewer 1 Report (Previous Reviewer 2)
Comments and Suggestions for Authors
All my comments have been addressed.
Author Response
Reviewer 1
1) All my comments have been addressed.
Reply: Thank you for the review.
Reviewer 2 Report (Previous Reviewer 3)
Comments and Suggestions for Authors
Hepatocellular carcinoma (HCC) is the most common primary malignancy of the liver, and it is a life-threatening disease of the liver. Patients who also have a blockage of the portal vein, which takes blood into the liver, are at particular risk for death, so any treatment modality which can improve the long-term outcome is of highest importance. The paper is there relevant to be published in Cancer/MDPI.
The retrospective investigation included 100 cases of males, but only 16 cases of females.
It is known that HCC is more common in males than in female, but is there any theory to why this is the case? In that case, it should be mentioned.
In the discussion, it would be beneficial for the reader if there would be a plan for future work using PBT for treatment of HCC.
Author Response
Reviewer 2
1)The retrospective investigation included 100 cases of males, but only 16 cases of females.It is known that HCC is more common in males than in female, but is there any theory to why this is the case? In that case, it should be mentioned.
Reply: Thank you for pointing that out. As you pointed out, the proportion of males in this study was higher. Since there is no difference between men and women in patient selection criteria, it is difficult to explain why. The fact that many cases in this study were treated without insurance and the degree of progression of vascular invasion may account for the gender difference. (L189-195)
2) In the discussion, it would be beneficial for the reader if there would be a plan for future work using PBT for treatment of HCC.
Reply: Thank you for pointing that out. We added several studies using PBT for treatment of HCC. (L247-255)
This manuscript is a resubmission of an earlier submission. The following is a list of the peer review reports and author responses from that submission.
Round 1
Reviewer 1 Report
Comments and Suggestions for Authors
This is a single institution retrospective analysis of HCC patients with PVTT treated with PBT. The long-term outcomes were favorable despite patients have a significant burden of disease. Some patients received "radical" treatment including the primary tumor and PVTT while others received "palliative" treatment only to the PVTT. I commend the authors on performing this analysis and think that it could meaningfully add to the published literature supporting PBT for HCC. I do have several major concerns that are described below.
1) The terms "radical" and "palliative" are confusing. At times "non-radical" is used in place of "palliative", for example see Figure 2. "Radical" is a term not commonly used at least in manuscripts from the West; consider using the term "definitive" instead. It is not clear if "palliative" patients had active disease in the liver other than the PVTT; if there was no active liver disease and if only the PVTT was treated by PBT then would this not should constitute "radical/definitive" treatment?
2) The study methods section should be made more clear as it states that the primary tumor and PVTT was treated although does not distinguish any differences between "radical" and "palliative" patients. The methods section also states that 72.6 GyE in 22 fractions was the "basic dose fraction" although it is clear from other parts of the manuscript that various fractionation schedules were used and it should be made more clear what they were and why only one fractionation schedule was not chosen. Was there a difference in fractionation based on tumor location, size, etc?
3) Patients treated with "palliative" treatment had very poor OS and the authors state that this "appears acceptable compared to no treatment or use of molecular-targeted drugs alone". Why should PBT therefore be recommended for this patient population especially considering that it is expensive and is delivered over multiple weeks for patients who have a very short projected survival time? I do not agree that the PBT data presented in this study support its use over SBRT to treat PVTT alone; I ask that the authors respond to this and suggest that they modify their Discussion/Conclusions to make is much clearer whether they believe that their data strongly support the routine use of PBT for "palliative" intent over other treatments.
4) Table 3 would be improved by describing whether PVTT alone versus primary tumor + PVTT was targeted in each of the studies listed. The outcomes among "radical" and "palliative" patients in this study were very different and the Table fails to describe this by describing outcomes for all patients in this study together.
Author Response
1) The terms "radical" and "palliative" are confusing. At times "non-radical" is used in place of "palliative", for example see Figure 2. "Radical" is a term not commonly used at least in manuscripts from the West; consider using the term "definitive" instead. It is not clear if "palliative" patients had active disease in the liver other than the PVTT; if there was no active liver disease and if only the PVTT was treated by PBT then would this not should constitute "radical/definitive" treatment?
Reply: Thank you for the review. As you pointed out, we decided to use "definitive" instead of "radical". Moreover, we supplemented the definitions of "definitive" and "palliative" with methods (L73-78, 87-96).
2) The study methods section should be made more clear as it states that the primary tumor and PVTT was treated although does not distinguish any differences between "radical" and "palliative" patients. The methods section also states that 72.6 GyE in 22 fractions was the "basic dose fraction" although it is clear from other parts of the manuscript that various fractionation schedules were used and it should be made more clear what they were and why only one fractionation schedule was not chosen. Was there a difference in fractionation based on tumor location, size, etc?
Reply: The dose fractionation is determined by the location of the lesion. Basically, 72.6 GyE in 22 fractions is adopted when the tumor is located at the porta hepatis, but when the lesion is close to the gastrointestinal tract, the dose is reduced to 2 GyE/fraction in consideration of the tolerable dose of the gastrointestinal tract. In addition, the dose fractionation may be adjusted according to other factors such as outpatient or not, patient’s health is adequate or not. 60 GyE in 15 fractions is the second most used dose fractionation in palliative treatment (L90-96).
3) Patients treated with "palliative" treatment had very poor OS and the authors state that this "appears acceptable compared to no treatment or use of molecular-targeted drugs alone". Why should PBT therefore be recommended for this patient population especially considering that it is expensive and is delivered over multiple weeks for patients who have a very short projected survival time? I do not agree that the PBT data presented in this study support its use over SBRT to treat PVTT alone; I ask that the authors respond to this and suggest that they modify their Discussion/Conclusions to make is much clearer whether they believe that their data strongly support the routine use of PBT for "palliative" intent over other treatments.
Reply: Thank you for pointing that out. As you pointed out, long-term irradiation of palliative patients seems to be undesirable. Palliative patients are assumed to have lymph node metastases or distant metastases other than PVTT. Therefore, it may be effective to irradiate PVTT with the minimum necessary dose for a short period of time and link it to molecular targeted drugs. We added sentence in the Discussion and Conclusion (L237-241, 243-246).
4) Table 3 would be improved by describing whether PVTT alone versus primary tumor + PVTT was targeted in each of the studies listed. The outcomes among "radical" and "palliative" patients in this study were very different and the Table fails to describe this by describing outcomes for all patients in this study together.
Reply: Thank you for pointing that out. In the other studies in Table 3, generally all cases are "definitive". Some of the studies include "palliative" cases with distant metastasis, but the target and treatment strategy are unknown. We added the number of distant metastasis cases in Table 3(Table 3).
Reviewer 2 Report
Comments and Suggestions for Authors
Too many patients were lost to follow-up in Figure 1 (local control). Therefore, they found the 5-year overall survival rate (18.0%) was significantly lower than the 5-year local control rate (86.1%). Tumor progression is the main cause of death.
Author Response
1) Too many patients were lost to follow-up in Figure 1 (local control). Therefore, they found the 5-year overall survival rate (18.0%) was significantly lower than the 5-year local control rate (86.1%). Tumor progression is the main cause of death.
Reply: Thank you for pointing that out. In past reports, most deaths after proton therapy for hepatocellular carcinoma were caused by recurrence from outside irradiated field. This study involves advanced cancer, and it is assumed that the ratio is even higher. The combination with molecular targeted drugs or immune checkpoint inhibitors may be effective in improving this situation. Similarly, Reviwer1 pointed out, we added sentence in the Discussion(L237-241).
Reviewer 3 Report
Comments and Suggestions for Authors
General comments
Hepatocellular carcinoma (HCC) is a serious and life-threatening disease of the liver. Patients 15 who also have a blockage of the portal vein, which takes blood into the liver, are at particular risk for death. The ms describes proton beam therapy for patients with HCC carcinoma with portal vein tumor thrombosis. To improve the treatment of HCC is therefor important. Since the paper describes that proton beam therapy can improve both the local control and overall survival rate, it is therefore of interest for the cancer community.
Specific comments
Title
I suggest that the title should be changed to “Proton beam therapy for treating patients with hepatocellular carcinoma with portalvein tumor thrombosis: a single center retrospective study”.
Introduction
Lines 42-44: The sentence “Since the 1970s, HCC has spread beyond an Eastern Asian predominance and 42 has increased in the Northern hemisphere, especially in the United States and Western Europe” is, without ref., copied word by word from Y. Ahmed et al., Review of hepatocellulat carcinoma: Epidimiology, etiology, and carcinogenesis, Journal of carcinogenesis, 2017. DOI: 10.4103/jcar.JCar_9_16.
I might be beneficial for the reader to add that Hepatocellular carcinoma (HCC) most often occurs in people with chronic liver diseases, such as cirrhosis caused by hepatitis B or hepatitis C infection. HCC is therefore predominant in Asian countries, including China, Mongolia, Southeast Asia, and Sub-Saharan Western and Eastern Africa (1).
Materials and methods
Line 67: Why was 100 cases male and only 16 cases female? Some explanation of that should be added.
Discussion
Of the 116 patients, 31 had received other therapy prior to PBT, and 63 received PBT as radical treatment, and 53 patients received PBT as palliative treatment, bit in the discussion only radical and non-radical cases are separated.
In the discussion, it is not clear if there is any significant difference between the 31 patients who had received other therapy prior to PBT, compared to patients who had not received any other therapy prior to PBT, within the 53 patients who received PBT as palliative treatment. The group is of course too small for any clear conclusion, but it would be of value to mention this anyway. Was there any difference in the results between men and woman?
Conclusion
The conclusion could be extended a little.
Author contributions are missing
Acknowledgements is missing
Comments on the Quality of English LanguageThe quality of English is satisfactory.
Author Response
General comments
Hepatocellular carcinoma (HCC) is a serious and life-threatening disease of the liver. Patients 15 who also have a blockage of the portal vein, which takes blood into the liver, are at particular risk for death. The ms describes proton beam therapy for patients with HCC carcinoma with portal vein tumor thrombosis. To improve the treatment of HCC is therefor important. Since the paper describes that proton beam therapy can improve both the local control and overall survival rate, it is therefore of interest for the cancer community.
Reply: Thank you for the review.
Specific comments
Title
I suggest that the title should be changed to “Proton beam therapy for treating patients with hepatocellular carcinoma with portal vein tumor thrombosis: a single center retrospective study”.
Reply: Thank you for your suggestion. We changed the title that way(L2-4).
Introduction
Lines 42-44: The sentence “Since the 1970s, HCC has spread beyond an Eastern Asian predominance and 42 has increased in the Northern hemisphere, especially in the United States and Western Europe” is, without ref., copied word by word from Y. Ahmed et al., Review of hepatocellulat carcinoma: Epidimiology, etiology, and carcinogenesis, Journal of carcinogenesis, 2017. DOI: 10.4103/jcar.JCar_9_16.
I might be beneficial for the reader to add that Hepatocellular carcinoma (HCC) most often occurs in people with chronic liver diseases, such as cirrhosis caused by hepatitis B or hepatitis C infection. HCC is therefore predominant in Asian countries, including China, Mongolia, Southeast Asia, and Sub-Saharan Western and Eastern Africa (1).
Reply: Thank you for pointing that out. We added the aforementioned literature in the References and sentence about the background of the HCC in the Introduction (L42-46, 265-266).
Materials and methods
Line 67: Why was 100 cases male and only 16 cases female? Some explanation of that should be added.
Reply: Thank you for pointing that out. HCC is more common in males, with a male to female ratio of 1:2.4 in the worldwide. We didn't arbitrarily select patients. The Japan Liver Cancer Association summarizes pathological progression with HCC by gender and shows that the proportion of males increases as the degree of progression in vascular invasion. We added sentence in the Discussion(L195-198).
Discussion
Of the 116 patients, 31 had received other therapy prior to PBT, and 63 received PBT as radical treatment, and 53 patients received PBT as palliative treatment, bit in the discussion only radical and non-radical cases are separated.
In the discussion, it is not clear if there is any significant difference between the 31 patients who had received other therapy prior to PBT, compared to patients who had not received any other therapy prior to PBT, within the 53 patients who received PBT as palliative treatment. The group is of course too small for any clear conclusion, but it would be of value to mention this anyway. Was there any difference in the results between men and woman?
Reply: Thank you for pointing that out. As Reviewer1 pointed out, we decided to use "definitive" instead of "radical" and supplemented the definitions of "definitive" and "palliative" with methods (L L73-78, 87-96). So, Prior treatment history does not mean the patient is palliative. Even in cases where previous lesions have been controlled long term by surgery or RFA and new lesions with PVTT have appeared, they are defined as "definitive" if the entire lesion is included in the irradiated area. There were no significant differences between men and women. The same was true for the medical history.
Conclusion
The conclusion could be extended a little.
Reply: We added a few sentences and revised the Conclusion (L243-246).
Author contributions are missing
Reply: We described Author contributions(L248-252).
Acknowledgements is missing
Reply: We described Acknowledgements(L260).
Round 2
Reviewer 1 Report
Comments and Suggestions for Authors
The authors have responded satisfactorily to my prior comments.
One additional minor point of clarification:
-Page 9, Line 230: What does "nodal" effect on the prognosis mean?